# Relationship among Agroclimatic Variables, Soil and Leaves Nutrient Status with the Yield and Main Composition of Kaffir Lime (*Citrus hystrix* DC) Leaves Essential Oil

**DOI:** 10.3390/metabo11050260

**Published:** 2021-04-22

**Authors:** Darda Efendi, Rahmat Budiarto, Roedhy Poerwanto, Edi Santosa, Andria Agusta

**Affiliations:** 1Department of Agronomy and Horticulture, Faculty of Agriculture, IPB University, Dramaga Bogor 16680, West Java, Indonesia; roedhy8@yahoo.co.id (R.P.); edisang@gmail.com (E.S.); 2Center for Tropical Horticulture Studies, IPB University, Dramaga Bogor 16680, West Java, Indonesia; 3Department of Agronomy, Faculty of Agriculture, Universitas Padjadjaran, Jatinangor Sumedang 45363, West Java, Indonesia; 4Study Program of Agronomy and Horticulture, Graduate School of IPB University, Dramaga Bogor 16680, West Java, Indonesia; 5Research Center for Chemistry, Indonesian Institute of Science, Puspitek Serpong, Tangerang Selatan 15314, Banten, Indonesia; bislunatin@yahoo.com or

**Keywords:** caryophyllene, citronellal, citronellol, linalool, rainfall, soil organic carbon, soil pH

## Abstract

Previous studies revealed the impact growing location has on the quantity and quality of essential oils derived from numerous *Citrus* spp., except on the kaffir lime. This study aims to analyze the relationship shared by agroclimatic variables and soil-plant nutrient status to kaffir lime leaves essential oil yield and main composition. The experiment was conducted between February and April 2019 in four growing locations, namely Bogor (6°36′36″ S, 106°46′47″ E), West Bandung (6°48′12″ S, 107°39′16″ E), Pasuruan (7°45′5″ S, 112°40′6″ E) and Tulungagung (8°6′27″ S, 112°0′35″ E). The highest essential oil yield was obtained from Bogor (1.5%), while the lowest one was from Tulungagung (0.78%). The yield was positively and significantly correlated with the rainfall, soil organic carbon, soil pH, and macronutrient levels, i.e., nitrogen, phosphorus, and magnesium. Citronellal, the major component in metabolites’ profile of kaffir lime leaves essential oils, was significantly affected by the growing location. The absolute content of citronellal was positively and significantly correlated with the actual soil pH and leaf Ca content; furthermore, it negatively correlated with the leaf content of Fe, Mn, Zn, Cu. Pearson correlation analysis also showed (i) a negative significant correlation between the relative percentage of citronellol and annual rainfall intensity; (ii) a negative significant correlation between altitude and relative percentage of caryophyllene, and (iii) a positive significant correlation between the relative percentage of linalool and leaf K content.

## 1. Introduction

Citrus essential oils proved to have many pharmacological properties, i.e., antioxidant [1], anticancer [2], anti-inflammatory [3], antiparasitic [4], antifungal [5], antibacterial [6], antimicrobial [7], larvacidal [8], stress release, and sleep [9]. One *Citrus* spp., kaffir lime (*Citrus hystrix* DC) is also rich in essential oil both in leaves and fruit [10] with numerous biological activities, such as anti-acne [11], anticancer [12], biolarvacidal [13], antimicrobial [14], and mood enhancer [15].

In Indonesia, kaffir lime essential oil is produced by the people of Tulungagung district in the context of product diversification of (fresh) kaffir lime leaves (KLL) which are widely cultivated [10]. In most Asian countries, including Indonesia, the leaf is mainly used as a cooking spice [16]. As the only production center of kaffir lime in Indonesia, the gate price of KLL essential oil in Tulungagung was IDR 700,000 per kg or less than 50 USD [10]. The price of this essential oil in the online market store has reached more than 150 USD per kg. The higher gap of price is associated with low-quality monitoring of oil produced. However, the demand for this product is still high, along with the development of the fragrance, cosmetics, and pharmaceutical industries; however, due to limited raw materials it has not been able to be fulfilled [10].

The increased production of raw materials for kaffir lime essential oil can be pursued through both extensification and intensification programs. The main principle of extensification is the cropland expansion, while the core of intensification is the improvement of input, i.e., plant materials, mechanization, fertilizer, irrigation, and pest control [17]. The extensification of kaffir lime should be arranged based on a deep understanding of how climate and soil impact the yield and quality of the product. However, in Asia, there is a tendency to transition from extensification to intensification due to the lack of cropland under a high density of population [18]. In the case of intensification, the determination of addressed input should be directed to the essential oil orientation and specific to location, since the determination of the best growing location is one of the agricultural innovations required in natural product development [19].

Growing locations, including geographic, soil and climate, along with cultural practices, especially water and nutrient management, were proved to highly affect the yield and quality of essential oil produced [19,20,21,22,23,24]. Macro-environmental factors such as the condition of the soil and monthly and annual climate modify the production of plant essential oil [25,26,27]. It is important to understand that climatic conditions during the growing period can affect the yield and composition of essential oils [28,29]. Monthly climatic conditions observed in every growing season can be used as the basis for crop irrigation arrangements and ultimately affect the production of plant essential oils [30,31]. A previous study by [32] proved that different seasons produce different yields and composition of essential oil. Evaluation of the seasonal climatic effect on the essential oil of rose-scented geranium reported that rainy and autumn support better plant growth and higher oil yield [28]. Moreover, the difference in annual climate has also been shown to impact the essential oils of two lamiaceae species [29]. Numerous studies revealed the effect of growing location elements on *Citrus* spp. essential oils, viz. *C. aurantium* L [33]; *C. aurantifolia* Swingle and *C. reticulata* Blanco [34]; *C. maxima* Burm, *C. paradisi* Macfad, *C. tangerina* Hort. Ex Tanaka, *C. limettioides* Tanaka, *C. latifolia* Tan, *C. limon* Tournef, *C. limonia* Osbeck and *C. sinensis* L. [35]. Unfortunately, there is still limited study related to the yield and quality of kaffir lime essential oils in response to growing locations. Therefore, this study aims to analyze the relationship among agroclimatic variables, soil-plant nutrient status to the KLL essential oil yield and main composition.

## 2. Results and Discussion

### 2.1. Altitude, Rainfall and Temperature of Kaffir Lime Growing Locations

Kaffir lime leaves were harvested from four different growing locations. Based on the altitude (Table 1), Bogor and Tulungagung were categorized as lowland, while Pasuruan and West Bandung were categorized as highland. Altitude displayed a strong, significant, and negative correlation to temperature, either average temperature, minimum temperature, or maximum temperature (Table 2). The increasing altitude was globally associated with the decrease in air temperature [36,37]. The average temperature in Bogor and Tulungagung were similar, while West Bandung recorded the lowest (Table 3) because of the highest altitude (Table 1). Rainfall intensity, both annual and monthly, were not correlated with the altitude. The highest rainfall, whether monthly or annually, was found in Bogor, while Tulungagung showed the opposite result.

### 2.2. Soil pH and Nutrient Status of Kaffir Lime Growing Locations

This study also revealed the variation of edaphic factors (soil pH and nutrient status) in the four locations. Soil actual pH (pH H_2_O) from Bogor and Pasuruan were relatively neutral (6–7) and significantly higher than those from Tulungagung and West Bandung (Table 4). West Bandung soil was consistently more acidic. Soil potential pH showed a significant and positive correlation (α 1%) to the soil actual pH, soil organic carbon (C-organic) and soil nitrogen total (N total) (Table 5). The highest pH in the Bogor soil was associated with the highest C-organic and N total. This finding highlighted the role of soil organic matter as a buffer to soil pH, leading to preserve soil C and N pool [38].

### 2.3. Plant Nutrient Status of Kaffir Lime

The present study displayed a significantly different result among four locations on plant macronutrient status, except for C-organic (Table 6). The N total of KLL from Bogor and Pasuruan was significantly higher than samples from West Bandung and Tulungagung. Based on the plant nutrient adequacy standard for fruit-oriented citrus [39], the N total from Bogor and Pasuruan was classified as optimum, whereas West Bandung and Tulungagung were categorized as low. The N status on the citrus plant should be well monitored since the N plays an important role for citrus plant, i.e., enhance vegetative growth, increase tree productivity, and modulate citrus fruit sugar and organic acid content [40,41]. The leaf N content showed strong (r 0.8), significant (α 1%) and positive correlation to the leaf phosphorus (P) content (Table 7).

The P total of KLL from West Bandung and Tulungagung was the lowest, while it was the highest for Bogor, followed by Pasuruan sample. Even though the West Bandung soil had the highest P (Table 4), its leaf P content was the lowest (Table 6), assumed to the incidence of less available P for the plant due to the acidic soil. In contrast, the relative neutral soil pH condition in Bogor supported the adequate absorption of P. The P is an important element and is needed for the formation of nucleic acid, phospholipid components, and ATP [42]. This finding also depicted the positive, strong (r 0.98) and significant (α 1%) correlation between leaf P total and leaf magnesium (Mg) content (Table 7). The Mg could stimulate the absorption and transport of P nutrient in plants [43]. The Mg sufficiency should be monitored in citrus plant [44], because this element is involved in numerous processes, i.e., carbohydrates metabolism, enzyme activation and protein synthesis [45]. The deficiency of Mg in citrus was previously reported to associate with the high uptake of potassium (K) [46]. Our finding accentuated a similar result where leaf Mg displayed the negative, strong (r 0.8) and significant (α 1%) correlation to leaf K (Table 7). The K^+^ ion was a prerequisite for plants for the process of sugar translocation, starch formation, stomatal opening regulation, and fruit quality improvement [47].

The calcium (Ca) content of KLL was significantly affected by locations, but no significant correlation to agroclimatic and edaphic variables. KLL sample of low land (Tulungagung and Bogor) had a significantly higher sulfur (S) than sample of high land (Pasuruan and West Bandung). However, a previous study reported an insignificant increase of S uptake as the increase of altitude from 1487 to 2569 m asl [48]. Both Ca and S were important essential secondary elements that regulate the citrus growth and development [49], i.e., (i) Ca for chromosome stability, cell division, cell elongation, second messenger, and cofactors of several enzymes associated with the hydrolysis of ATP and phospholipids [49,50]; (ii) S for constructing of many vitamins, protein, chlorophyll and some phytohormones [49,51].

The present work revealed there was a significant effect of growing locations on micronutrients status of KLL (Table 8). Leaves harvested from West Bandung had the highest levels of iron (Fe), manganese (Mn), copper (Cu) and zinc (Zn). This could be associated with the acidic character of the West Bandung soil (Table 4). The high solubility and plant uptake of micronutrient is often reported in acid soil [52,53]. Our work also reported there was a negative, strong, and significant (α 1%) correlation between certain micronutrients, i.e., Fe, Mn and Zn to leaf N content.

### 2.4. Essential Oil Yield of Kaffir Lime Leaves

Bogor kaffir lime leaves produced the highest yield of essential oils (Figure 1). The lowest yield was noticed in KLL from Tulungagung and West Bandung. The yield showed a positive, strong (r 0.83) and significant (α 1%) correlation to annual rainfall of the locations (Table 9). The higher intensity of rainfall in Bogor supported kaffir lime to have greater essential oil production. Water sufficiency was thought to have a beneficial effect on essential oil production of kaffir lime. It was reported that water insufficient conditions decreased the essential oil yield of *Dracocephalum moldavica* [30], *Lavandula latifolia* and *Salvia sclarea* [31]. Additionally, there was a reduction in the essential oil yield of *Pelargonium graveolens* during summer [28].

The essential oil yield was also influenced by edaphic factors since there was a positive significant correlation between the yield to soil C-organic status (r 0.77) and soil pH (r 0.78) (Table 10). The highest content of soil C-organic found in Bogor stimulated the kaffir lime to have the highest essential oil production. Soil organic matter improved soil microflora [54], soil water retention, release, and storage of plant nutrients [55]. The level of soil organic matter was affected by pH, relative humidity, temperature, texture, microbial population, oxygen availability, and the management of soil [56]. Success stories of the addition of manure to enhance essential oil yield were previously reported in *Allium cepa* L. [57], *Coriandrum sativum* L. [58], *Melisa officinalis* L. [59], *Mentha piperita* L. [60], *Ocimum basilicum* [61], *Origanum vulgare* L. [62], and *Pogostemon cablin* [63].

Plant nutrient status also affected the yield of KLL essential oil. Plant nutrient levels of N, P and Mg showed a strong (r 0.93, r 0.86 and r 0.92, respectively), positive, and significant (α 1%) correlation to the oil yield (Table 11). The N and Mg were two important macronutrients for the production of chlorophyll in citrus plants [47]; N-deficient leaves displayed smaller chloroplasts [64]. A chloroplast is a form of plastid organelle, which becomes the biosynthesis site of monoterpene [21], the largest constituent of KLL essential oil [65]. It was assumed that the sufficient level of N and Mg was a supporting factor for increasing oil yield. Earlier reports confirmed the increase of essential oil yield as the increase of N fertilization in dragonhead [30], basil [66,67], chamomile [68], and rosemary [69]. Previous study reported the higher (31%) oil yield obtained from oregano treated with Mg and Ca through foliar feeding [70], while P fertilization increased oil yield in basil [71] and *Pelargonium graveolens* [72,73].

### 2.5. Main Composition of KLL Essential Oil

There were four major components frequently reported regarding KLL essential oil, namely citronellal, citronellol, linalool [16,74,75,76,77], and caryophyllene [74,78]. Citronellal is an economically important compound due to its use as an intermediate component in the synthesis of perfumes, drugs and basic ingredients for the synthesis of isopulegol, menthol and citronellol [79]. The pharmacological properties of this monoterpenoid aldehyde were anti-inflammatory [80,81] and antifungal activities [82]. Citronellal was the most dominant compound found in metabolites fingerprinting of KLL essential oils, with the relative concentration ranging from 61.7–74.8% [75,76,77,83,84]. Our quantitative analysis approach revealed the absolute concentration of citronellal by using a mathematical equation (R^2^ 0.9975) (Figure 2a). The absolute content of citronellal was significantly affected by the location (Figure 2b). The order from the highest to the lowest citronellal yield was Pasuruan > Bogor = Tulungagung > West Bandung.

In earlier studies, citronellal relative content in two cultivars of *Pelargonium graveolens* was affected by the growing season, where the rainy season produced a higher result than during summer [32]. In contrast, our finding showed a positive, strong (r 0.81) significant (α 1%) correlation between the citronellal content and soil actual pH, instead of climatic variables. Soil from Pasuruan has an actual pH that is closer to neutral (6.27) compared to acidic soil from West Bandung (4.76). Citrus generally grows optimally in slightly acidic to neutral soil due to the availability of more balanced macro and micronutrients. Macronutrients, specifically leaf Ca content showed a positive, significant and strong (r 0.83) correlation to the citronellal content. Leaf micronutrient, i.e., Fe, Mn, Cu, Zn, correlated significantly and negatively with citronellal content. The levels of leaf Mn and Fe in this experiment were proved to have negatively correlated with leaf N level (r −0.85 and r −0.80, respectively). However, the mechanism of the Ca, Fe, Mn, Cu, and Zn into citronellal biosynthesis in kaffir lime is still unclear.

Growing locations significantly affected the relative content of citronellol of KLL essential oil (Figure 3). Our work revealed that the citronellol content in four locations varied from 2.42% to 6.55%, with the highest coming from Tulungagung, while the lowest was from Pasuruan. Previous studies in different sites reported that the citronellol content in KLL essential oil was derived from Selangor, Melaka, and Terengganu, at 6.6%, 6.7% and 13.4%, respectively [75,76,77]. Earlier studies report the success of fertilizer application to enhance the content of citronellol in rose-scented geranium essential oil, viz. phosphate solubilizing bacteria (PSB) biofertilizer by 16.6% [73] and NPK fertilizer + poultry manure by 26% [72]. Citronellol, as a natural monoterpenoid alcohol, could act as a material for mosquito repellent [85] and, as such, showed anti-inflammatory [80,81] and antifungal [82] activities.

Another monoterpenoid alcohol noticed was linalool with its pharmacological properties such as antidiabetic [86,87], antifungal [82], antibacterial [88,89] and pest control agent [90]. The present work revealed the content of linalool in KLL essential oil by about 2.8–4.5%. The lowest relative percentage of linalool was found in the oil sample from Pasuruan, whereas the highest was from the West Bandung (Figure 3). Linalool content in earlier studies on Malaysia KLL varied from 1–3.9% [75,76,77]. The present work also highlighted that the relative level of linalool was positively correlated with the K nutrient content of KLL (Table 10). Similar to the present finding, a previous study by [91] reported that the administration of K fertilizer could be used to enhance the content of linalool. Harvesting at rainy season also proved to increase linalool content in rose-scented geranium essential oil [32]. The application of vermicompost proved to increase the linalool content by about 109% in the lemongrass (*Cymbopogon citratus*) [92].

Caryophyllene is a sesquiterpene hydrocarbon compound found in *Citrus* peel and leaves [77,93,94] that has long been used as a flavor enhancer and fragrance [95]. This compound possessed antimicrobial, anti-inflammatory, anticancer, antioxidant and antibiotic activities [80,81,96]. The present work found varying relative percentages from 0–2.55% in the essential oil of KLL (Figure 3). A previous study by [77] reported that the caryophyllene content in essential leaves of *Citrus hystrix* and *Citrus x microcarpa* was 0.9% and 2.8%, respectively. This study also noticed that the increased of altitude was likely associated with a decrease in the relative levels of caryophyllene. An earlier report stated that the intensification culture practice in the form of NPK fertilizer, vermicompost, and PSB biofertilizer could increase the caryophyllene relative content by about 24%, 180% and 124%, respectively [92]. The variation of the main constituent and yield of kaffir lime essential oil in the present experiment was confirmed the earlier hypothesis that essential oil yield and composition in aromatic plants may be varied in response to genotype, growing location and culture practices [65,72,91,97].

## 3. Materials and Methods

### 3.1. Study Site

The experiment was conducted between February and April 2019. Distillation and analysis of essential oil quality were carried out at the Indonesian Spice and Aromatic Plant Research Institute (Balitro), Indonesia. The analysis of gas chromatography–mass spectrometry (GCMS) to profile metabolite and quantify citronellal was conducted at the DKI Jakarta Provincial Health Laboratory, Indonesia. Soil and leaves nutrient status was analyzed in the laboratory of the Department of Agronomy and Horticulture, Faculty of Agriculture, IPB University, Indonesia.

### 3.2. Plant Material Preparation

Plant materials in the form of kaffir lime leaves were harvested from four different locations in Java, namely Bogor, West Bandung, Tulungagung and Pasuruan (Table 1) in March during the rainy season. Data on altitude, average temperature, minimum temperature, maximum temperature, monthly rainfall, and annual rainfall were downloaded from an online library search [98,99,100,101]. Monthly and annual climatic data collection is relatively more accurate in describing land conditions and their effect to plant growth, especially for an annual plant, such as kaffir lime. The oldest and youngest fully developed leaves harvested in each tree were around 1-year-old and 1-month-old, respectively. Tulungagung KLL were harvested from 1.5-year-old plants that were planted in the monoculture garden owned by the local community. West Bandung KLL were harvested from 2-year-old plants, which are intercropped with leafy vegetables in the community’s vegetable garden. Pasuruan KLL were harvested from 2.5-year-old plants that are planted in the residents’ backyard. Bogor KLL were harvested from 1.5-year-old plants arranged in the monoculture system at Pasir Kuda IPB experimental station.

About 3 kg of KLL were harvested in every location and were immediately packaged in a closed container equipped with a cooling gel and then sent to the laboratory within the same day. Samples were withered for two days before distillation. Water-steam distillation was carried out in every kilogram of KLL by using a stainless steel distillery kettle for about 3 h. Essential oil was dried with anhydrous sodium sulfate and then the volume was measured, after which the oil was stored in dark glass bottles for further testing purposes.

### 3.3. Procedure of Essential Oil Analysis

The essential oil observed variables were oil yield, citronellal content, the relative content of citronellol, linalool and caryophyllene. The yield of essential oil was calculated from the ratio between the weight of essential oil against the fresh weight of KLL (1 kg) before being distilled and expressed in % (*v*/*w*).

Determination of citronellal content and relative content of citronellol, linalool and caryophyllene was carried out using the instrument of Agilent 7890 gas chromatography- tandem to 5975 mass spectrometry (Agilent Technologies Inc., Santa Clara, CA, USA), with 1 µL injection volume in the split mode (a split ratio 10:1). The column used was Agilent 19091N-133HP-INNOWax Polyethylene Glyco with dimensions of size 30 m (length) × 250 μm (diameter) × 0.25 μm (thickness). The carrier gas used was helium (mobile phase) with a constant flow mode and a flow velocity of 0.9 µL per minute and a column pressure of 7.06 psi. The column temperature was programmed from 60 °C to 210 °C, with an increased stage of (i) 2 °C per minute to 150 °C and then maintained for 1 min at 150 °C; and (ii) 20 °C per minute to 210 °C and then maintained for 10 min at 210°C. The injector temperature during the analysis was programmed constant at 230 °C.

There were two types of samples injected, standard citronellal compound and KLL essential oil. The standard compound was used to make regression curves for estimating citronellal concentration in essential oils. Regression curves were composed of 10 concentrations of standard citronellal compound, i.e., 1, 2, 3, 4, 5, 6, 7, 8, 9 and 10%. To prepare 1% standard solution, about 10 μL of standard citronellal solution was needed and then ethyl acetate (990 μL) was added to form 1 mL of the final solution. Citronellal samples in 1 mL vials were then vortexed for forming homogeneous samples and then injected into GCMS. The GCMS analysis resulted in the peak area of the citronellal compound (X) and then regressed with the level of concentration of standard citronellal solution (Y). Regression equation in the standard curve could be used to estimate absolute citronellal content in essential oils of KLL.

Essential oil samples were taken as much as 100 μL and then ethyl acetate (900 μL) added to form a 1 mL sample solution (10 × dilution factor). There were three replications for every growing location. The sample was then vortexed to form a homogeneous solution before GCMS injection. The peak area of citronellal produced by GCMS was used as mathematical input in the regression equation that has been produced to calculate the absolute citronellal content in the kaffir lime essential oil samples. Also, the results of GCMS analysis could also be used to determine relative levels of other compounds. The relative content of citronellol, linalool and caryophyllene was determined based on these compounds’ peak area expressed in percent of the total area of the chromatogram peak.

### 3.4. Data Analysis

Soil and leaves nutrient status were analyzed for each sampling location in triplicate. Soil nutrient analysis included the Walkley and Black method for the C-organic (%), the pH meter method for pH H_2_O and pH KCl, the Kjeldahl method for the N total (%), ultraviolet-visible (UV-VIS) spectrophotometer for the total P (mg P_2_O_5_ 100 g sample^−1^) and atomic absorption spectrometry (AAS) for the total K (mg K_2_O 100 g sample^−1^). Leaf nutrient analysis included the Walkley and Black methods for the C-organic (%), the Kjeldahl method for the N total (%), the UV-VIS spectrophotometer for the total P (%) and S (%), as well as AAS for the K total (%), Mg (%), Ca (%), Na (%), Fe (ppm), Mn (ppm), Cu (ppm) and Zn (ppm).

All data obtained was analyzed for Pearson correlation. Certain data, i.e., oil yield, citronellal content, soil and leaves nutrient status was analyzed by analysis of variance and then continued by Duncan’s multiple range test (DMRT) at α 5% level for any significant difference found. Statistical analysis was done using Statistical Tool for Agricultural Research (STAR) version 2.0.1.

## 4. Conclusions

Four different growing locations with variations in terms of altitude, agroclimatic conditions (temperature and rainfall), soil nutrient status (pH, C-organic, N, P, K) and leaves nutrient status (C-organic, N, P, K, Ca, Mg, S, Fe, Mn, Cu, Zn) clearly influenced the yield and main composition of the essential oils of KLL. The yield was positively correlated with rainfall, C-organic status, soil pH and leaves nutrient levels of N, P, and Mg. As the main constituent of KLL essential oils, the absolute content of citronellal was positively correlated with soil actual pH and leaves Ca nutrient content. However, it was negatively correlated with Fe, Mn, Zn and Cu content in leaves. Other important constituents showed various results, i.e., (i) citronellol displayed a negative correlation to annual rainfall intensity, (ii) the increase of altitude was associated with the decrease of relative content of caryophyllene, and (iii) relative content of linalool was positively correlated with the K content of KLL.

## Figures and Tables

**Figure 1 metabolites-11-00260-f001:**
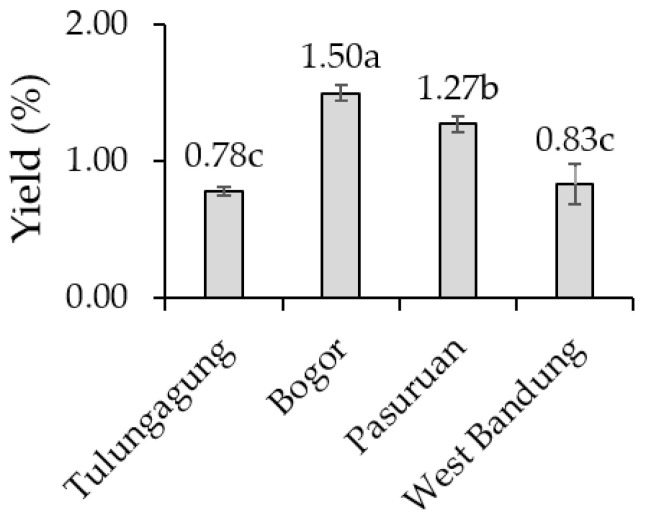
The essential oil yield of kaffir lime leaves in four growing locations. Note: in the bar chart of yield, the different alphabet above the rectangular bar is significantly different based on the Duncan’s multiple range test at α 5%; the error bar represents the standard deviation.

**Figure 2 metabolites-11-00260-f002:**
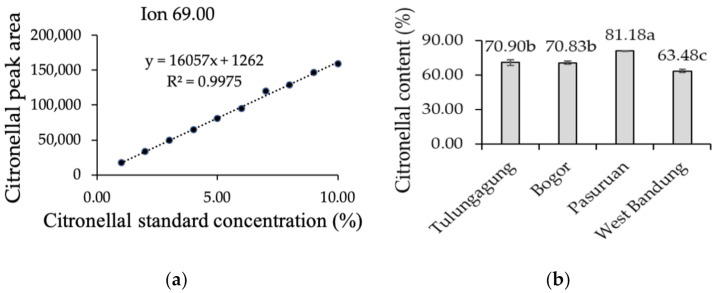
(**a**) Optimization curves of citronellal content in the ion 69; (**b**) Citronellal content in the kaffir lime leaves essential oils from four growing locations. Note: in the bar chart of citronellal content, the different alphabet above the rectangular bar is significantly different based on the Duncan’s multiple range test at α 5%; the error bar represents the standard deviation.

**Figure 3 metabolites-11-00260-f003:**
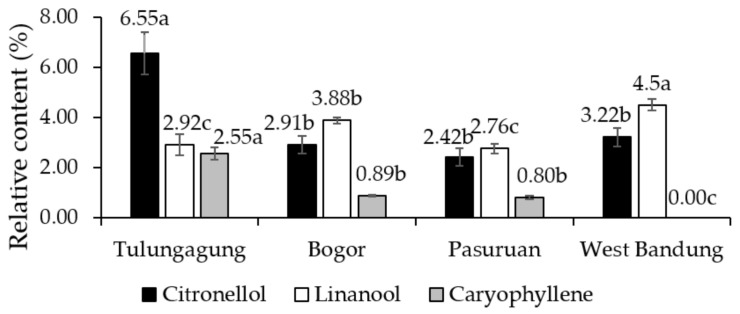
The relative content of citronellol, linalool, and caryophyllene in the GCMS profile of kaffir lime leaves essential oils from four growing locations. Note: the different alphabet above the similar color of rectangular bar is significantly different based on the Duncan’s multiple range test at α 5%; the error bar represents the standard deviation.

**Table 1 metabolites-11-00260-t001:** Sampling locations for kaffir lime leaves.

Sampling Locations (Village, District)	Locations’ Coordinates (Latitude, Longitude, Altitude)
Ngunut, Tulungagung	8°6′27″ S, 112°0′35″ E, 109 m asl
Pasirkuda, Bogor	6°36′36″ S, 106°46′47″ E, 239 m asl
Jatiarjo, Pasuruan	7°45′5″ S, 112°40′6″ E, 803 m asl
Cikole, West Bandung	6°48′12″ S, 107°39′16″ E, 1189 m asl

**Table 2 metabolites-11-00260-t002:** Pearson coefficient correlation of several agroclimatic variables such as altitude, temperature and rainfall in four kaffir lime growing locations.

Variables	Al	AR	MR	Tav	Tmin
AR	0.17				
MR	0.17	1			
Tav	−0.98 **	−0.04	−0.04		
Tmin	−0.95 **	0.01	0.01	0.98 **	
Tmax	−0.92 **	−0.07	−0.07	0.90 **	0.87 **

Note: **—significantly correlated at 99% confidence level. Al—altitude, Tav—average temperature, Tmin—minimum temperature, Tmax—maximum temperature, MR—monthly rainfall, AR—annual rainfall.

**Table 3 metabolites-11-00260-t003:** Temperature and rainfall of four kaffir lime growing locations.

Locations	Tav (°C)	Tmin (°C)	Tmax (°C)	MR (mm)	AR (mm)
Tulungagung	25.3	20.1	30.1	145	1743
Bogor	25.4	20.1	30.1	342	4104
Pasuruan	22.1	17.4	26.9	261	3126
West Bandung	20.3	15	24.1	250	3001

Note: Tav—average temperature, Tmin—minimum temperature, Tmax—maximum temperature, MR—monthly rainfall, AR—annual rainfall.

**Table 4 metabolites-11-00260-t004:** Macronutrient status (C-organic, N, P, K) and pH values of soil samples under the kaffir lime canopy in four locations.

Locations	pH H_2_O	pH KCL	C-Organic (%)	N Total (%)	P Total (mg P_2_O_5_ 100 g^−1^)	K Total (mg K_2_O 100 g^−1^)
Tulungagung	5.72 ± 0.04 b	4.87 ± 0.08 c	1.49 ± 0.02 b	0.2 ± 0.01 b	139.03 ± 1.20 c	34.96 ± 3.96 b
Bogor	6.24 ± 0.11 a	5.66 ± 0.22 a	2.37 ± 0.02 a	0.27 ± 0.01 a	242.23 ± 18.45 b	163.60 ± 13.56 a
Pasuruan	6.27 ± 0.04 a	5.24 ± 0.12 b	1.46 ± 0.06 b	0.19 ± 0.01 b	136.36 ± 9.46 c	38.21 ± 4.62 b
West Bandung	4.76 ± 0.09 c	3.89 ± 0.10 d	0.97 ± 0.05 c	0.16 ± 0.01 c	352.25 ± 17.72 a	166.57 ± 10.51 a

Note: Numbers in the same column followed by the same letter are not significantly different based on Duncan’s multiple range test at α 5% level.

**Table 5 metabolites-11-00260-t005:** Pearson coefficient correlation of several soil macronutrients and pH under kaffir lime canopy in four growing locations.

Variables	C-Organic	N Total	P Total	K Total	pH H_2_O
N total	0.98 **				
P total	−0.25	−0.23			
K total	0.18	0.19	0.90 **		
pH H_2_O	0.75	0.72	−0.73	−0.40	
pH KCl	0.88 **	0.86 **	−0.61	−0.22	0.97 **

Note: **—significantly correlated at 99% confidence level.

**Table 6 metabolites-11-00260-t006:** Macronutrient status of kaffir lime leaves grown in four locations.

Locations	C-Organic (%)	N Total (%)	P Total (%)	K Total (%)	Mg (%)	Ca (%)	S (%)
Tulungagung	40.36 ± 0.61 a	1.83 ± 0.06 b	0.14 ± 0.00 c	2.18 ± 0.09 a	0.17 ± 0.01 d	2.70 ± 0.28 b	0.38 ± 0.02 a
Bogor	41.97 ± 1.18 a	2.5 ± 0.06 a	0.25 ± 0.01 a	1.64 ± 0.05 d	0.39 ± 0.02 a	2.12 ± 0.30 c	0.42 ± 0.17 a
Pasuruan	40.22 ± 2.10 a	2.4 ± 0.09 a	0.17 ± 0.01 b	1.99 ± 0.03 b	0.26 ± 0.01 b	3.29 ± 0.14 a	0.15 ± 0.02 b
West Bandung	40.94 ± 1.23 a	1.72 ± 0.01 b	0.15 ± 0.02 c	1.85 ± 0.06 c	0.20 ± 0.01 c	1.82 ± 0.36 c	0.34 ± 0.12 ab

Note: Numbers in the same column followed by the same letter are not significantly different based on Duncan’s multiple range test at α 5% level.

**Table 7 metabolites-11-00260-t007:** Pearson coefficient correlation of macro- and micronutrients status of kaffir lime leaves in four growing locations.

Leaves Nutrient	C	N	P	K	Mg	Ca	Na	S	Fe	Mn	Cu
N	0.18										
P	0.40	0.80 **									
K	−0.44	−0.48	−0.78								
Mg	0.37	0.83 **	0.98 **	−0.80 **							
Ca	−0.44	0.36	−0.17	0.46	−0.14						
Na	0.44	0.61	0.88 **	−0.70	0.88 **	−0.31					
S	0.18	−0.14	0.23	−0.32	0.15	−0.37	0.41				
Fe	−0.11	−0.80 **	−0.52	−0.02	−0.53	−0.56	−0.45	0.20			
Mn	−0.05	−0.85 **	−0.52	0.00	−0.53	−0.64	−0.38	0.28	0.98 **		
Cu	−0.07	−0.64	−0.19	−0.21	−0.22	−0.70	−0.06	0.45	0.81 **	0.84 **	
Zn	0.00	−0.87 **	−0.51	0.16	−0.55	−0.58	−0.24	0.55	0.78	0.87 **	0.75

Note: **—significantly correlated at 99% confidence level.

**Table 8 metabolites-11-00260-t008:** Micronutrient status of kaffir lime leaves grown in four locations.

Locations	Fe (ppm)	Mn (ppm)	Cu (ppm)	Zn (ppm)	Na (%)
Tulungagung	375.96 ± 116.35 b	57.41 ± 7.17 b	2.21 ± 0.23 b	23.83 ± 1.35 a	0.003 ± 0.001 b
Bogor	9.30 ± 8.47 c	23.96 ± 1.65 c	2.08 ± 1.89 b	14.63 ± 2.07 b	0.008 ± 0.001 a
Pasuruan	4.38 ± 2.48 c	7.74 ± 1.07 d	0.09 ± 0.01 c	7.49 ± 0.40 c	0.003 ± 0.001 b
West Bandung	1497.33 ± 282.94 a	133.81 ± 10.11 a	4.57 ± 0.89 a	26.37 ± 2.11 a	0.003 ± 0.001 b

Note: Numbers in the same column followed by the same letter are not significantly different based on Duncan’s multiple range test at α 5% level.

**Table 9 metabolites-11-00260-t009:** Pearson correlation coefficient of kaffir lime leaves essential oil (yield and main composition) and geo-climatic factor (altitude, temperature, and rainfall).

Variables	Y	Clal	Clol	Llol	Car
Al	−0.22	−0.20	−0.58	0.47	−0.83 **
AR	0.83 **	0.06	−0.85 **	0.44	−0.67
MR	0.83 **	0.06	−0.49	0.26	−0.39
Tav	0.33	0.17	0.49	−0.39	0.75
Tmin	0.41	0.25	0.41	−0.41	0.70
Tmax	0.29	0.28	0.45	−0.46	0.73

Note: **—significantly correlated at 99% confidence level. Y—yield of oil, Al—altitude, AR—annual rainfall, MR—monthly rainfall, Tav—average temperature, Tmin—minimum temperature, Tmax—maximum temperature, Clal—citronellal content, Clol—relative levels of citronellol, Llol—relative levels of linalool, Car—relative levels of caryophyllene.

**Table 10 metabolites-11-00260-t010:** Pearson correlation coefficient of kaffir lime leaves essential oil (yield and main composition) and edaphic factor (soil pH and soil macronutrient status).

Variables	Y	Clal	Clol	Llol	Car
C-org	0.77 **	0.25	−0.13	−0.11	0.21
N	0.73	0.20	−0.08	−0.09	0.25
P	−0.13	−0.41	−0.35	0.98 **	−0.73
K	0.23	−0.69	−0.45	0.95 **	−0.68
pH H_2_O	0.73	0.81 **	−0.17	−0.67	0.34
pH KCl	0.78 **	0.65	−0.15	−0.51	0.33

Note: **—significantly correlated at 99% confidence level. Y—yield of oil, C-org—carbon organic, N—nitrogen total, P—phosphate total, K—potassium total, Clal—citronellal content, Clol—relative levels of citronellol, Llol—relative levels of linalool, Car—relative levels of caryophyllene.

**Table 11 metabolites-11-00260-t011:** Pearson correlation coefficient of kaffir lime leaves essential oil (yield and main composition) and plant macro- and micro-nutrient status.

Variable	Y	Clal	Clol	Llol	Car
C-org	0.30	−0.16	−0.18	0.33	−0.17
N	0.93 **	0.66	−0.58	−0.30	−0.12
P	0.86 **	0.14	−0.49	0.18	−0.21
K	−0.67	0.26	0.69	−0.66	0.66
Mg	0.92 **	0.20	−0.57	0.18	−0.28
Ca	0.18	0.83 **	0.04	−0.88 **	0.39
Na	0.71	−0.07	−0.24	0.26	−0.05
S	0.00	−0.63	0.33	0.41	0.17
Fe	−0.66	−0.80 **	0.05	0.70	−0.45
Mn	−0.70	−0.87 **	0.17	0.72	−0.34
Cu	−0.45	−0.89 **	0.14	0.75	−0.29
Zn	−0.74	−0.88 **	0.58	0.53	0.11

Note: **—significantly correlated at 99% confidence level. Y—yield of oil, C-org—carbon organic, N—nitrogen total, P—phosphate total, K—potassium total, Clal—citronellal content, Clol—relative levels of citronellol, Llol—relative levels of linalool, Car—relative levels of caryophyllene.

## Data Availability

Data available on request due to privacy restriction.

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
