# Peer review of "Relationship among Agroclimatic Variables, Soil and Leaves Nutrient Status with the Yield and Main Composition of Kaffir Lime (Citrus hystrix DC) Leaves Essential Oil"

_metabolites, 2021, doi:10.3390/metabo11050260_

Round 1
Reviewer 1 Report
This manuscript is dedicated to a study of the correlation between the agroclimatic variables, soil and leaves nutrients and the yield and composition of the essential oil of kaffir lime Citrus hystrix from Indonesia. The methods applied are up-to-date and appropriate. The conclusions are supported by the results. The results are of interest. There are however several concerns, as follows:
- The article is too long, discussions often too wordy. Should be re-written in a more concise way.
- In Figures 1, 2b, and 3 standard deviations must be presented.
- In Tables 4, 6, 8 standard deviations must be presented.
- Subsection 4.4. Data Analysis: the first sentence (in lines 374 – 375) does not belong to this subsection, it should be placed before the subsection 4.2. Plant material preparation.
- A radical improvement of the English language is mandatory.
Reviewer 2 Report
The manuscript by Darda Efendi et al. deals with the relationship among agroclimatic, soil and leaves status with the yield and composition of essential oils of kaffir lime leaves.
The topic might be interesting but it is insufficient for publication in a journal “Metabolites”. The reason of my negative evaluation is only one point but it is critical.
The metabolites in leaves drastically change in response to environmental stimuli, but authors evaluated only annual or monthly climatic factors. It is necessary to evaluate daily climate changes during one or two weeks before harvesting the experimental materials. The result and discussion might change after evaluation the relationship between daily climate factors and essential oil status. Authors should reanalyze and rediscuss the results and then, resubmit the manuscript.
Reviewer 3 Report
The presented manuscript under the title of Relationship among agroclimatic variables, soil and leaves nutrient status with the yield and main composition of kaffir lime (Citrus hystrix DC) leaves essential oil’ is well designed and informative. I have no issue regarding the manuscript and the final decision I will leave for the academic editor.
Round 2
Reviewer 1 Report
The article has been significantly improved.
Reviewer 2 Report
The manuscript was carefully revised, and the results and discussion are valid. Thus, this paper should be acceptable for publication in Metabolites.